# Improvement in Healthy Meal Index, Lunch Quality, and Diversity Scores Following an Integrated Nutritional Intervention in a Communal Dining Room: The NEKST Study

**DOI:** 10.3390/nu12061741

**Published:** 2020-06-10

**Authors:** Ofira Katz-Shufan, Tzahit Simon-Tuval, Liron Sabag, Danit R. Shahar

**Affiliations:** 1The S. Daniel Abraham International Center for Health and Nutrition, Department of Public Health, Faculty of Health Sciences, Ben-Gurion University of the Negev, Beer Sheva 8410501, Israel; lironsharon1@gmail.com (L.S.); dshahar@bgu.ac.il (D.R.S.); 2Department of Health Systems Management, Guilford Glazer Faculty of Business and Management and Faculty of Health Sciences, Ben-Gurion University of the Negev, Beer Sheva 8410501, Israel; simont@bgu.ac.il

**Keywords:** catering service, integrated nutritional intervention, diet diversity, diet quality, simple healthy meal index, fruits and vegetables

## Abstract

The objective of the NEKST (Nutrition Environmental Kibbutzim Study), a nonrandomized interventional study, was to evaluate the effect of an integrated intervention program on participants’ lunch quality and diversity in two communal dining rooms (intervention *n* = 58 vs. control *n* = 54). The intervention included recipe modification, environmental changes, and an education program. The outcomes included simple healthy meal index (SHMI), lunch quality (LQS), and diversity scores (LDS) calculated based on photographs of lunch trays. A nutrition questionnaire assessed the changes in fruit and vegetable intake at baseline and 3 months following the intervention. The mean SHMI, LQS, and LDS increased in the intervention group (0.51, *p* < 0.001; 0.27, *p* = 0.045; 0.95, *p* < 0.001, respectively) but not in the control group (*p* = 0.865; *p* = 0.339; *p* = 0.354, respectively). Multivariable linear models demonstrate an increase in the SHMI (β = 0.26, 95% CI [0.12–0.76], *p* = 0.015), LQS (β = 0.23, 95% CI [0.06–0.83], *p* = 0.024), and LDS (β = 0.34, 95% CI [0.41–1.39], *p* < 0.001) of the participants in the intervention group. More participants in the intervention group raised their daily fruit intake compared with the control. We conclude that this integrated intervention program was effective in improving lunch healthy meal index, quality, and diversity in a communal dining room, with a modest halo effect of the intervention throughout the day.

## 1. Introduction

Nutritional quality is a cornerstone in preventing chronic diseases and maintaining health [1]. One of its key expressions is the increased intake of desirable food groups such as fruits, vegetables, whole grains, nuts, and legumes, and limited intake of saturated and trans fats, added sugars, and sodium [2,3]. These components, particularly the intake of fruits and vegetables, have consistently been shown to be associated with health outcomes [4,5]. An increased diversity of foods within healthy diet components helps to ensure an adequate intake of essential nutrients and to promote health [6]. Therefore, focusing on improving diet quality and diversity and increasing fruit and vegetable intake through an intervention program may be beneficial in improving public health.

The settings-based approach to health promotion adopts intervention strategies targeted at minimizing disease risk factors by creating a supportive and enabling environment [7,8]. One example of a food-related environment is catering services, which in many cases serve low-nutritional quality foods, resulting in an unhealthy environment [9,10,11,12]. Alternatively, catering services as an environmental setting may provide a promising opportunity to promote a healthier diet [13,14,15] by addressing a large group of diners simultaneously.

As suggested in the NOURISHING framework, a good policy to promote a healthier diet should contain three domains of action: the food system, the food environment, and behavior change communications [16]. In previous works, changes in the food served, such as increasing the availability and variety of healthy food options, enhanced diet quality [17,18,19,20,21,22], and environmental actions, such as changing the location of different food items on a buffet [23] or communicating nutritional messages in the food environment [24,25,26], encouraged adopting a healthier diet. Using a combination of these domains of action has been shown to have a positive effect on nutritional choices [27,28,29,30,31,32,33,34,35]. However, in most studies, intervention programs were based solely on dietary recommendations rather than on diners’ preferences, and study outcomes were based on cashier sales [24,26,28,32,36,37], and self-reported food choices using participant interviews and/or questionnaires [27,28,29,32,34]. Several recent intervention studies in dining systems, mainly at military bases [22,30,31], examined the effect of interventions on food consumption using the digital food photography method, yet they did not follow the same cohort of participants before and after the intervention [22,30,31,33].

In the current study, an intervention program was developed based on all three domains of action. In contrast with the extant literature [22,30,31,33], we evaluated its impact by following a cohort of participants and providing robust evidence with regard to their diet quality and diversity. In addition, in contrast with the extant knowledge thus far [22,30,31], we provide evidence from a non-military setting.

## 2. Methods

### 2.1. Study Design and Procedure

The NEKST (Nutrition Environmental Kibbutzim Study) is a nonrandomized interventional study that was conducted in two large catering systems of two kibbutzim (kibbutzes) that share similar characteristics: Kibbutz Magen (intervention group) and Kibbutz Nir-Yitzhak (control group). A kibbutz is a collective, a uniquely Israeli way of living. It is a small community with several hundred members, and its livelihood is derived from agriculture and industry [38]. One characteristic of the kibbutz is its communal dining room in which all members, their families, and factory workers from the region have their meals together.

The participants who agreed to participate in the study and met the inclusion criteria were recruited following obtaining their written informed consent. In addition, after choosing food for lunch, the participants were asked to enable us to take pictures of their food tray on three randomly selected days during one week before and after the intervention. All the outcome measures were collected before the intervention (baseline) and three months following it (Time 3). The Ben-Gurion University Human Subjects Research Committee approved the study protocol (#1397-1).

### 2.2. Study Population

A convenience sample of diners was recruited during February and March 2017. Kibbutz members and factory workers who eat in the dining room of Kibbutz Magen or Nir-Yitzhak were asked to voluntarily participate in the study through an advert that was hung in central locations and sent to their email. The ad informed participants about the study’s focus on healthy lifestyle and nutrition. The diners were invited to an introductory meeting where the research project and the intervention program were explained in detail. They were recruited either upon completion of that meeting, in the dining room, or at the kibbutz factories. The inclusion criteria were regular diners 30+ years of age who ate lunch at least three times a week in the kibbutz dining room and who choose their food themselves. We recruited 77 participants in the intervention group and 76 in the control group, of whom 67 in each group completed the study (10 participants from the intervention group and 9 from the control group were lost to follow-up). The photos of lunch trays were obtained for 58 participants in the intervention group and 54 in the control group. A flowchart of participant enrollment, follow-up, and those who were included in the data analysis is shown in Figure 1.

### 2.3. The Intervention Program

The intervention program was developed to promote healthy food choices at lunch in a communal dining room. It was based on elements of previous programs that were found to be effective [16] and was adjusted to diners’ preferences, as derived from a qualitative inquiry using in-depth interviews among 13 diners who ate in kibbutz dining rooms on a regular basis. These interviewees were not recruited during the current study. The interviews included questions about the diners’ dietary habits, food choices, and attitudes towards various intervention programs.

The intervention program included the following components:

**Nutritional changes,** including the nutritional improvement of some recipes according to dietary recommendations toward a healthier diet, mainly reduced sodium, sugar, and saturated fat.

**Environmental changes in the dining room,** including changes in the presentation location of some dishes and labeling healthy dishes with a green “Like” based on the recommendations of the Israeli Ministry of Health Regulation Committee to promote healthy nutrition [39].

**Health communications** in an ongoing campaign to encourage healthier food choices using an electronic board and posters in the dining room.

**An education program** that was the single component administered to both the intervention and control groups. Three educational lectures were conducted during the intervention period: two concerning healthy nutrition by a registered dietitian and one encouraging physical activity by a physical activity specialist. The participants who did not attend the lectures received summaries of them by email.

### 2.4. Study Measures

The main outcome measures were the simple healthy meal index (SHMI), lunch quality score (LQS), lunch diversity score (LDS), self-reported daily consumption of fruits and vegetables, and the preferred choice of something sweet (e.g., candy, sweetened yogurt, energy bar, or fruit). The photographs of diners’ lunch trays were analyzed by direct visual estimation, similarly to previous work with the validated novel remote food photography method (RFPM) [40,41]. For this analysis, a nutritionist who was acquainted with the dishes served in the two dining rooms identified and counted the food groups in each dish. For all three scores, the Choices Food Profiling Criteria [42] were used to assign each food into the appropriate food group. The identification of the categories in each food group as healthy/unhealthy were based on the definition of healthy food in the new Israeli Ministry of Health guidelines for the Israeli population [43]. The photographs of the lunch trays were taken before participants began eating; thus, the identification referred only to the food choices and not to the food actually consumed.

#### 2.4.1. Simple Healthy Meal Index

The validated SHMI was calculated based on three key components: the number of fruit and vegetable portions (excluding potatoes), the number of fat portions relative to the number of starch portions and the source of these fats (mainly animal or mainly vegetable-based), and the number of whole grain and potato portions [44]. The indexing was as follows: fruits and vegetables (0 = less than 1 unit, 1 = minimum 1 unit, 2 = minimum 2 units); fat content and quality (0 = more fat units than starchy units; 1 = same number of fat and starchy units, with fat mainly animal-based; 2 = fewer fat units than starchy units or the same number of fat and starchy units, with fat mainly vegetable-based); whole grain and potatoes (0 = less than 0.5 unit, 1 = minimum 0.5 unit, 2 = minimum 1 unit). The SHMI was the sum of the points given for each component, and the index values ranged from 0 to 6. A mean score was calculated for each participant based on his/her three lunch tray photographs at baseline and after three months (Time 3). Cronbach’s alpha of the SHMI at baseline was 0.702, and 0.623 at Time 3.

#### 2.4.2. Lunch Quality Score

The LQS was calculated following previous work [45] and based on the presence of healthy categories of each food group in the lunch tray. Seven food groups were addressed in this score: main dish (rich in protein), carbohydrate side dish, vegetables, pulses, other side dishes (e.g., soup, sauces, bread), desserts (fruit or snacks), and drinks. Dishes in each food group received a score of 1 for those identified as healthy or 0 for those identified as unhealthy. The terms for scoring were: main dish (0 = processed, 1 = prepared from fresh ingredients), carbohydrate side dish (0 = refined grains, 1 = whole grains), vegetables (0 = fried vegetables or 0–2 vegetables, 1 = 3 or more fresh vegetables), pulses (0 = no pulses, 1 = pulses), other side dishes (0 = high-sugar, high-fat, and high-sodium items, 1 = low-sugar, low-fat, low-sodium items), desserts (0 = sugary sweets, 1 = fruit or fresh fruit juice), and drinks (0 = all sugar-sweetened and artificially-sweetened beverages, 1 = water or soda water). A missing food category received a score of zero. The sum of the points given for each food category was used as the LQS, with scores ranging from 0 to 7. A mean score was calculated for each participant based on his/her three lunch tray photographs at baseline and at Time 3. Cronbach’s alpha of the LQS at baseline was 0.671 and 0.736 at Time 3.

#### 2.4.3. Lunch Diversity Score

The LDS was estimated based on a validated score [46]. One point was assigned for the presence of any amount of food from each of the following food groups: protein (meat, meat products, soya products, fish, dairy products, or eggs) and carbohydrate (cereals, pasta, rice, potatoes, or bread). In the case of vegetables (including pulses) and fruits, 1 point was assigned for each different fruit or vegetable. The scores ranged from 0 to 13. A mean score was calculated for each participant based on his/her three lunch tray photographs at baseline and at Time 3. Cronbach’s alpha of the LDS at baseline was 0.845 and 0.888 at Time 3.

For each lunch tray, the correlation coefficient between the SHMI and LQS was 0.326 (*p* < 0.001, *n* = 601), between the SHMI and LDS 0.256 (*p* < 0.001, *n* = 601), and between the LQS and LDS 0.561 (*p* < 0.001, *n* = 601). These moderate correlations justified analyzing the change in each score following the intervention in order to provide robust evidence of the effectiveness of the intervention program. Figure 2 presents examples of lunch tray photographs and their corresponding SHMI, LQS, and LDS.

Self-Reported intake of daily fruit and vegetable was estimated by the Diet Quality Questionnaire [47].The number of fruit and vegetable portions that were consumed daily by the participants was received from his/her answer to the questions: “How many vegetable servings are you eating per day?” and “How many servings of fruit are you eating per day?”, with four possible answers of none, 1–2 servings, 3–4 servings, or 5 or more servings.

The preferred choice for something sweet was evaluated by the question: “When you feel like having something sweet, what do you usually prefer?”. The possible answers were: a) candy, cookies, cake, or chocolate; b) sweetened yogurt; c) an energy bar; d) fruit. This variable was analyzed as a dichotomous one that equaled 0 if a participant reported a, b, or c and 1 if a participant reported d (the choice of fruit).

The participants’ characteristics were collected at baseline by the researchers. These included age, gender, marital status, employment status, level of education, place of residency, and self-reported adoption of physical activity. Anthropometric measurements were measured in light clothing and without shoes. Weight was measured using digital scales with an accuracy of 0.1 kg (Tanitatype). Height was measured using a folding portable altimeter with accuracy of 0.1 cm (Secatype). These measures were used to calculate the diners’ body mass index (BMI) in order to compare participants in the intervention and control group at baseline and adjust the multivariable analyses to this characteristic. We assessed the participants’ nutritional and physical activity knowledge using a questionnaire based on Parmenter’s General Nutrition Knowledge Questionnaire for adults [48].

### 2.5. Statistical Analyses

Data analysis was carried out using IBM SPSS Statistics software (version 23, Armonk, NY, USA). To examine the impact of the intervention on between-groups and within-group differences, we used the Mann–Whitney and Wilcoxon tests, respectively. For dichotomous variables, we analyzed between-groups and within-group differences using x^2^ and a McNemar’s test, respectively. For the exploration of the change in the SHMI, LQS, and LDS following the intervention, we specified multivariable linear models. The independent variable of interest was the study group (intervention vs. control) and other explanatory variables that had significant bivariate associations and satisfied *p* ≤ 0.1 in the multivariable analyses, including living in the kibbutz, education level, and the nutritional and physical activity knowledge score. The statistical significance level was set at 0.05.

## 3. Results

Our analysis included 58 participants in the intervention group and 54 participants in the control group. At baseline, 70 participants had three lunch tray photos, 30 had two, and 12 had one. After the intervention (Time 3), 82 participants had three lunch tray photos, 23 had two, and 7 had one. As shown in Table 1, both groups were comparable at baseline with regard to age (*p* = 0.232), gender (*p* = 0.823), educational level (*p* = 0.057), % living in the kibbutz (*p* = 0.151), BMI (*p* = 0.355), nutritional and physical activity knowledge score (*p* = 0.207), and physical activity performance (*p* = 0.287). Since the educational level had borderline significance, we adjusted all the multivariable models to this variable.

### 3.1. Change in the Simple Healthy Meal Index Following the Intervention

The change in the SHMI is presented in Table 2. This index increased by 0.51 in the intervention group (*p* < 0.001) but not in the control group (*p* = 0.734), and the between-groups difference was significant (*p* = 0.003). This increase was influenced by the increase in the fruit and vegetable component of the score (*p* < 0.001). There was no difference between the fat, whole grain, and potato components of the score (*p* = 0.145, *p* = 0.206, respectively) in the intervention group. In terms of the actual portions (rather than a relative score), there was an average increase of 0.71 (±0.98) in the fruit and vegetable portions at lunch in the intervention group (*p* < 0.001), compared to an insignificant increase of 0.15 (±0.78) in the number of these portions in the control group (*p* = 0.168). A significant difference was observed between the groups in this regard (*p* = 0.001). A multivariable linear analysis, presented in Table 3a, revealed that the SHMI of participants in the intervention group increased more following the intervention compared to the control group (β = 0.26, 95% CI (0.12–0.76]), *p* = 0.015).

### 3.2. Change in Lunch Quality Score Following the Intervention

As presented in Table 2, the mean LQS increased by 0.27 in the intervention group (*p* = 0.045) but not in the control group (*p* = 0.283). The difference between groups was significant (*p* = 0.029). This increase stemmed predominantly from an increase in vegetables (*p* = 0.004) and healthy desserts (fruit) (*p* < 0.001). However, the score of the other side dishes decreased significantly following the intervention (*p* = 0.040). These trends were not observed in the control group (Table 2). In addition, in the control group, we observed a decrease in carbohydrate side dishes (*p* = 0.002) and in unsweetened drinks (water or soda water) (*p* = 0.019). The only significant increase in the components of the quality score in this group was with regard to pulses (*p* = 0.010); however, this did not influence the total LQS (Table 2). A multivariable linear analysis (Table 3b) revealed that the LQS of participants in the intervention group increased more following the intervention compared to the control group (β = 0.23, 95% CI (0.06–0.83), *p* = 0.024).

### 3.3. Change in Lunch Diversity Score Following the Intervention

Table 2 presents the changes in the LDS. It reveals that the mean LDS increased by 0.95 in the intervention group (*p* < 0.001) but not in the control group (*p* = 0.464), with a significant difference between groups (*p* = 0.001). This increase stemmed predominantly from the increase in vegetable (*p* < 0.001) and fruit (*p* < 0.001) components of the score. These trends were not observed in the control group (*p* = 0.269 and *p* = 0.244, respectively). The multivariable linear regression that is presented in part c of Table 3 reveals that the LDS increased more in the intervention group compared to the control group (β = 0.34, 95% CI (0.41–1.39), *p* < 0.001). 

### 3.4. Self-Reported Daily Consumption of Fruits and Vegetables

Beyond the change in lunch choices, there was a significant improvement in the self-reported daily consumption of fruit portions following the intervention in the intervention group. A total of 20.4% of the participants increased their daily consumption and 2% decreased it (*p* = 0.008) following the intervention. In the control group, 14.9% of participants increased their daily fruit consumption following the intervention and 8.5% decreased it (*p* = 0.285). No difference was detected in the change in fruit intake between the groups (*p* = 0.238), and one should note that the groups were not comparable at baseline. Specifically, participants in the intervention program consumed less fruit (in gr) at baseline compared to the control group (146.4 ± 113.7 vs. 178.4 ± 95.5, respectively, *p* = 0.039). There was no improvement in the daily consumption of vegetables in either the control or the intervention group.

### 3.5. The Preferred Choice for Something Sweet

There was a significant improvement in the preferred choice for something sweet in the intervention group. Following the intervention, 14.3% of the participants changed their preferred choice of something sweet from a less healthy option to fruit, and no one changed their choice from fruit to a less healthy option (*p* = 0.007). In contrast, this trend was not observed in the control group, where 14.3% changed their choice of sweet food to fruit, and 4.9% changed their choice of sweet food from fruit to a less healthy option (*p* = 0.118). However, no significant difference in change was observed between the groups (*p* = 0.780).

## 4. Discussion

This study evaluated the effect of an integrated intervention program versus an educational program only on the food choices of regular diners in a communal dining room. We found that the intervention program was effective in improving the food quality and diversity of participants’ food choices during lunch, mainly due to an increase in the choice of fruits and vegetables. In addition, the intervention program was moderately effective in increasing the daily consumption of fruit and replacing the less healthy choice of something sweet with fruit.

We have not come across studies that examined simultaneously the three different scores (SHMI, LQS, and LDS) that reflect both the nutritional quality and diversity of lunch. These scores were based on the visual processing of photographs of lunch foods prepared by a catering service. The visual processing of photographs, as opposed to self-reported dietary intake, allows the objective assessment of a meal chosen by diners. Our results are analogous with other studies that evaluated integrated intervention programs in institutional dining using the visual processing of photographs to evaluate diet quality by the Healthy Eating Index (HEI) and food-group consumption [30]. Others also examined meal calories and nutrient intake [22,31,33], and showed a positive nutritional change. Contrary to our work, these studies did not examine the change in diet quality and diversity of a cohort of patients, but rather examined the dietary intake of independent groups of diners before and after interventions.

Although the change in daily intake of fruits was modest, the fact that the improvements in the SHMI, LQS, and LDS were significant and mainly derived from a preference shift towards fruit and vegetables may be an indication of better daily nutrition. Those results are in accord with those of another study indicating that the availability of fruit and vegetables at lunch led to a better dietary profile during the day [49]. These findings suggest that lunch-focused intervention may have a halo effect beyond lunch on participants’ daily diet and eating behavior. Fruits and vegetables are an important part of a healthy diet [50], and it has consistently been shown that a greater intake of these foods is associated with a lower risk of non-communicable diseases and all-cause mortality [5]. Thus, our evidence of the improved consumption of fruit and vegetables may lead to a reduction in the comorbidity burden and associated healthcare costs. Further study is warranted to estimate those potential benefits that may offset the costs of this multilevel intervention strategy.

Our study has several limitations. First, there was no randomization of assignment to the study groups. We chose two similar dining rooms with the same kibbutz characteristics, although there was a difference in the active involvement of the catering service staff and the kibbutz management. Second, the recruitment of a convenience sample may have led to an incomplete and inaccurate sampling frame and an inability to generalize the study findings due to a self-selection bias. Finally, our objective measures of the SHMI, LQS, and LDS were based on participants’ choices rather than on their actual consumption. Thus, further study should focus on complementary data on the quality and diversity of actual intake.

Notwithstanding these limitations, this study evaluated the effect of an integrated environmental nutritional intervention on the quality and diversity of diners’ food choices at lunch in a real-life setting using lunch tray photographs, an objective and innovative method, among a cohort of diners, thus enriching and extending current knowledge. In addition, the evaluation of the food selected focused on the quality and variety of the choice rather than on the quantity of nutrients. This goes along with the new nutritional recommendations published by the Israeli Ministry of Health [43] and is similar to international recommendations [51,52]. The intervention program that was applied in the current study and the corresponding results may contribute to the development of catering-level interventions and may serve as a model for similar environmental interventions in catering systems in various settings. Further research is needed to explore diners’ long-term compliance and satisfaction with the intervention program. Moreover, it would be valuable to test the program with catering systems in different settings and to evaluate whether and which specific parts of this strategy are cost effective.

## Figures and Tables

**Figure 1 nutrients-12-01741-f001:**
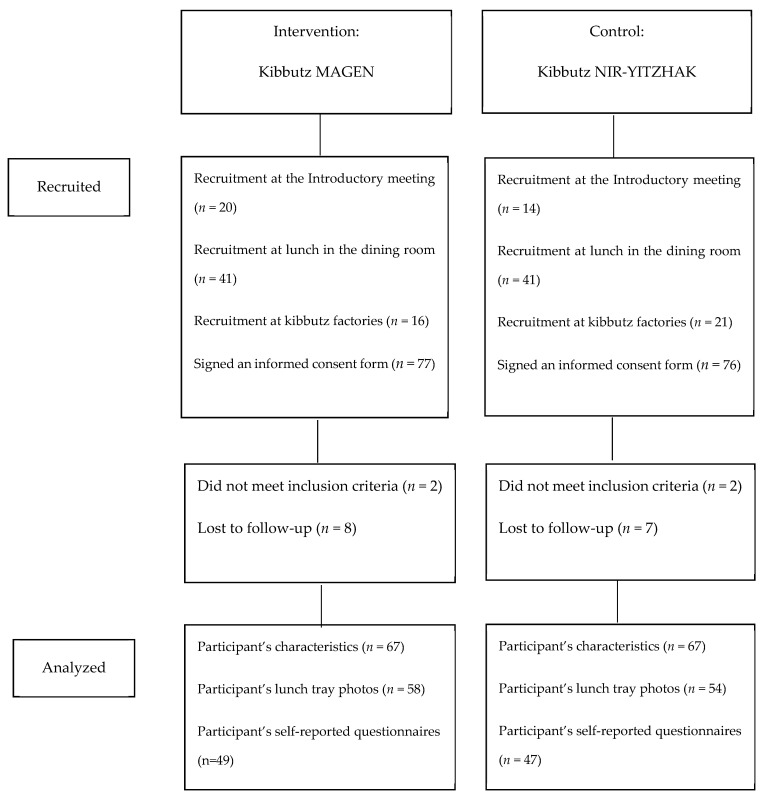
Flowchart of participant enrollment and follow-up.

**Figure 2 nutrients-12-01741-f002:**
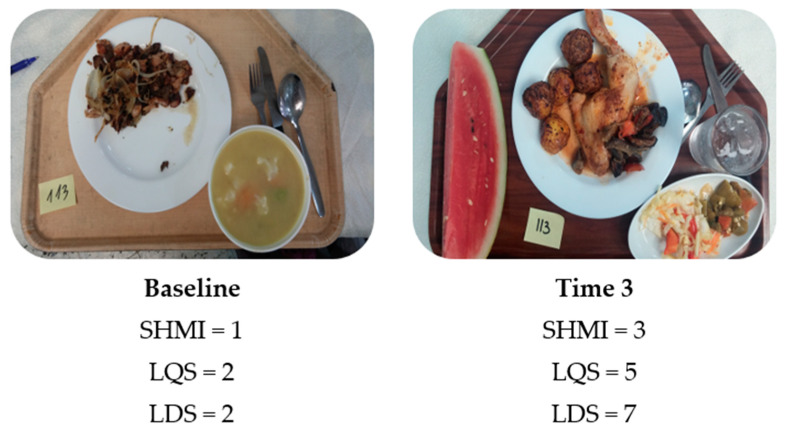
Example of the simple healthy meal index, quality, and diversity scores estimated based on the lunch tray photographs of participant #113 before and after the intervention.

**Table 1 nutrients-12-01741-t001:** Baseline characteristics of the participants by study group.

	Intervention Group (Magen)	Control Group (Nir Yitzhak)	
n	58	54	
Age (years) ^a^	56.8 ± 15.5 (59, 32–83)	61.5 ± 14.9 (60, 30–87)	0.232 ^b^
Gender—male ^c^	31 (53.4%)	30 (55.6%)	0.823 ^d^
Live in the kibbutz	50 (87.7%)	41 (77.4%)	0.151 ^d^
Educational level ^b^ Academic education	33 (61.6%)	21 (38.9%)	0.057 ^d^
Nutritional & physical activity knowledge score ^a, e^	22.9 ± 4.6 (24, 10.5–30)	22.0 ± 4.6 (21.5, 11–31)	0.207 ^b^
BMI (kg/m^2^) ^a^	28.7 ± 4.9 (28, 21–43)	27.8 ± 4.9 (27, 19–39)	0.355 ^b^
Performance of physical activity ^c, e^	43 (79.6%)	42 (87.5%)	0.287 ^d^

^a^ Values are mean ± SD (median, minimum–maximum). ^b^ Mann–Whitney rank sum test. ^c^ Values are n (%). ^d^ χ^2.^. ^e^ In these comparisons, *n* = 53 in the intervention group and *n* = 47 in the control group.

**Table 2 nutrients-12-01741-t002:** Changes in the simple healthy meal index (SHMI), lunch quality score (LQS), and lunch diversity score (LDS) following the intervention (Time 3 compared to baseline), by study group.

Lunch Score Followed by Its Components	Intervention Group (Magen), n = 58		Control Group (Nir Yitzhak), n = 54		*p*-Value between Groups ^c^
	Mean Change ^a^	*p*-Value ^b^	Mean Change ^a^	*p*-Value ^b^	
**SHMI**	0.51 ± 0.88	<0.001 ^d^	0.04 ± 0.76	0.734 ^d^	0.003 ^e^
Fruits and vegetables	0.26 ± 0.54	<0.001	0.04 ± 0.32	0.468	0.018
Fat	0.12 ± 0.49	0.145	0.07 ± 0.28	0.061	0.275
Whole grain and potatoes	0.13 ± 0.58	0.206	−0.07 ± 0.64	0.379	0.116
**LQS**	0.27 ± 1.01	0.045 ^d^	−0.15 ± 1.00	0.283 ^d^	0.029 ^e^
Main dish (rich in protein)	0.01 ± 0.41	0.832	−0.10 ± 0.36	0.051	0.171
Carbohydrate side dish	0.02 ± 0.39	0.828	−0.17 ± 0.37	0.002	0.022
Vegetables	0.19 ± 0.45	0.004	0.08 ± 0.31	0.091	0.058
Pulses	−0.03 ± 0.17	0.220	0.11 ± 0.27	0.010	0.001
Other side dishes	−0.09 ± 0.36	0.040	0.04 ± 0.28	0.247	0.023
Deserts	0.28 ± 0.49	<0.001	0.02 ± 0.14	0.317	<0.001
Drinks	−0.03 ± 0.38	0.681	−0.10 ± 0.30	0.019	0.236
**LDS**	0.95 ± 1.30	<0.001 ^d^	0.13 ± 1.27	0.464 ^d^	0.001 ^e^
Protein	0.01 ± 0.22	0.841	0.01 ± 0.24	0.844	0.981
Carbohydrate	−0.06 ± 0.36	0.169	−0.03 ± 0.36	0.348	0.682
Vegetables	0.80 ± 1.40	<0.001 ^d^	0.19 ± 1.24	0.269 ^d^	0.015 ^e^
Fruit	0.20 ± 0.36	<0.001	−0.03 ± 0.20	0.244	<0.001

Abbreviations: SHMI—simple healthy meal index; LQS—lunch quality score; LDS—lunch diversity score. ^a^ Values are mean ± SD. ^b^ Wilcoxon signed rank test. ^c^ Mann–Whitney rank sum test. ^d^ Paired sample *t*-test. ^e^ Independent sample *t*-test.

**Table 3 nutrients-12-01741-t003:** Multivariable linear models for the change in the outcome measures following the intervention.

(**a**): Determinants of change in simple healthy meal index.			
	**β**	**95% CI**	***p*-Value**
Study group (intervention vs. control)	0.26	0.12–0.76	0.015
Nutritional & physical activity knowledge score	−0.24	−0.08–(−0.01)	0.016
Education (academic vs. other)	0.08	−0.20–0.46	0.447
n	100		
Adjusted R^2^	0.091		
(**b**): Determinants of change in lunch quality score.			
	**β**	**95% CI**	***p*-Value**
Study group (intervention vs. control)	0.23	0.06–0.83	0.024
Live in the kibbutz (yes vs. no)	−0.22	−1.20–(−0.09)	0.024
Nutritional & physical activity knowledge score	−0.21	−0.09–(−0.003)	0.036
Education (academic vs. other)	−0.06	−0.50–0.28	0.584
n	98		
Adjusted R^2^	0.120		
(**c**): Determinants of change in lunch diversity score.			
	**β**	**β**	***p*-Value**
Study group (intervention vs. control)	0.34	0.41–1.39	<0.001
Education (academic vs. other)	−0.16	−0.90–0.07	0.094
n	112		
Adjusted R^2^	0.101		

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
