# Peer review of "Improvement in Healthy Meal Index, Lunch Quality, and Diversity Scores Following an Integrated Nutritional Intervention in a Communal Dining Room: The NEKST Study"

_nutrients, 2020, doi:10.3390/nu12061741_

Round 1
Reviewer 1 Report
Dear Authors, following are only a few suggestions for your manuscript.
Titel:
Exclude the words “Results from” to shorten the title.
Abstract:
Please include your specific design in your abstract.
Introduction:
Delete the author guideline information from lines 66-78.
Methods:
Study population: In this section you need to include the conducted sampling strategy (and not only in the limitation part) and more information about your recruitment process.
The intervention program: in line 128, please cite the programs, i.e. add a literature reference, if these programs are from the literature.
Reviewer 2 Report
Overall appreciation
It is quite interesting to study and to develop interventions to promote healthier eating habits and this study aimed to evaluate the effectiveness of an integrated intervention focused on 3 different domains based on previous evidence. However, this manuscript presents some methodological issues and the study design and procedure could be clearer. Furthermore it presents important limitations that could influence the conclusions drawn (e.g. results were based on participants’ choices rather than on their actual consumption). Moreover, there are several formatting errors and I have some specific questions presented in the minor comments below. The authors claim to have adjusted recipes to diner’s preferences (I presumed only in the intervention group) and it would be interesting in future research to evaluate if the integrated intervention regarding the 3 domains of action was responsible for the results or if it was this factor alone. Moreover some words and phrases have lost their meaning in translation so this article should be edited by a native English speaker.
Comments/Questions
Line 32 – Reference is missing
Lines 43, 44– Sentence doesn’t seem properly written
Line 58– What do you mean by: “did not follow the same cohort of participants, what cohort of participants are we talking about?
Line 60-65 – Should be in the methods section
Line 66 – Reference unformatted
Line 66 – 78 Paragraphs from the instructions for authors (???)
Line 90 – “They then filled out the study questionnaires”- sounds off - and questionnaires regarding what?
Line 93 – Sentence doesn’t seem properly written “All outcome measures were elicited before the intervention (baseline) and three months following it” - Change for collected or evaluated
Line 96 and 98 – Lack of coherence for the designation of criteria
Lines 105-123 - Flowchart of participant enrollment and follow-up refers things not mentioned before, what was the initiation conference?
Line 130 – I don´t fully understand how were diners’ preferences assessed, what questions did they answered and how were they selected? Why only 13? Are they included in the final sample? Are they from both groups?
Were diner’s preferences only considered in the intervention group? And only in this group were made changes in the recipes taking into account those preferences, if so it would be interesting to evaluate in future research if this factor alone was responsible for the results.
Line 139 – If you did this part of the intervention in the control group that means that the “control” group was also intervened and cannot be named as a control group. What you are doing is comparing two different interventions.
Line 145, 146 – References regarding validation of those tools are missing
Line 186 – similar to the diversity component of the quality score. What diversity component?
Line 197 – font seams different
Line 207 – The question doesn’t evaluate the preferred choice for between-meal snacks, it only refers to sweet ingredients…
Line 215 – The methodology used for height and weight measurement is missing
Line 247 - font seams different
Line 255 – “Changes in SHMI, LQS, and LDS following the intervention” what does following the interventions mean? The data collected at time3?
Tables 2 and 3 should be in the end of the description of the changes in all scores, otherwise it forces the reader to scroll up and down.
Line 273: font seams different;
Line 273- the category “other dishes” only refers to healthy food? Because when you explained the LQS tool in lines 178 and 179 it states that other side dishes includes both healthy and unhealthy options. So I don´t understand this sentence “ However, other dishes
defined as healthy (e.g., soup and sauces) decreased significantly following the intervention (p=0.040). – you only took into account the healthy ones from this category in your analysis?
Line 275-276 - font seams different; healthy carbohydrate additive- sounds off
292 – And were both groups comparable at baseline regarding the intake of fruit and vegetables?
Line 301-303 – I struggled at first to understand this sentence.
Lines 312-313 - Sentence doesn’t seem properly written
Lines 329-330 – This isn´t very accurate concerning that the influence on fruit intake was modest and the information on line 297-298
Lines 341-342 – This is a strong limitation because individuals in the intervention group could choose healthier options specifically adding more portions of fruits/vegetables to their plate because they know they are supposed to do it (and their plate is being photographed) but that doesn’t mean they ate it.
Line 368 – All references have the identification number twice.
Line 462 – Reference from Wikipedia? there is no such information in a better source?
Q1 - Why were height and weight measured? Just to compare BMI at baseline? This could be explained in the text.
Q2- Why was the education program implemented in both groups? To conclude that this intervention alone doesn’t present results? If that was the case, there is no conclusion on this matter.
